# Groundwater Circulation in the Xianshui River Fault Region: A Hydrogeochemical Study

**Yuqing Zhao [1,2,3], You-Kuan Zhang [4,\*], Yonglin Yang [5], Feifei Li [5] and Sa Xiao [4]**

[1]   Guangxi Key Laboratory of Environmental Pollution Control Theory and Technology,
    Guilin University of Technology, Guilin 541004, China; 2020163@glut.edu.cn
[2]   Collaborative Innovation Center for Water Pollution Control and Water Safety in Karst Area,
    Guilin University of Technology, Guilin 541004, China
[3]   Guangxi Key Laboratory of Environmental Pollution Control Theory and Technology for Science and
    Education Combined with Science and Technology Innovation Base, Guilin University of Technology,
    Guilin 541004, China
[4]   Guangdong Provincial Key Laboratory of Soil and Groundwater Pollution Control, Southern University of
    Science and Technology, Shenzhen 518055, China; 11849230@mail.sustech.edu.cn
[5]   Survey Engineering Institute, Sichuan Earthquake Administration, Ya'an 625099, China;
    ylyang@126.com (Y.Y.); lifeifei0114@sina.com (F.L.)
\*   Correspondence: zhangyk@sustech.edu.cn

**Abstract:** Water samples from rainfall, river, springs, and wells in the Xianshui River fault region near Xialatuo, China were collected during two sampling campaigns to investigate the complex groundwater circulation in the region. The major ions, stable isotopes, and four natural radium isotopes of the water samples were analyzed, and the results were utilized to identify different groundwater circulation depths. Most water samples excluding the one at a hot spring and the one at a borehole possess similar hydrochemical compositions and lower total dissolved solids (TDS), implying that their circulation depth is relatively shallow or that residence time is short. The sample at the hot spring has high TDS and high temperature as well as the high F concentration, inferring that it may circulate at a deeper depth. The sample at the borehole contains mixed hydrochemical characteristics of other samples. Three groundwater flow systems may exist in the study area: the shallow groundwater system recharged by precipitations and local groundwater flow, the deep groundwater system recharged by the regional groundwater flow, and the intermediate one between the above two systems. The finding of the three flow systems is supported by the $\delta^2$H and $\delta^{18}$O as well as the apparent radium ages of the samples. The $\delta^2$H and $\delta^{18}$O values at the intercept of the line formed by the shallow groundwater samples and the local meteoric water line (LMWL) are similar to those of modern precipitations. The $\delta^2$H and $\delta^{18}$O values at the intercept of the line formed by the deep groundwater samples and the LMWL show that it is probably recharged by relatively older precipitations. The $^2$H and $^{18}$O values of the borehole samples are between the above two intercept points. The deep-circulated groundwater with high temperature has longer apparent radium age than other water samples. The apparent radium ages of the shallow groundwater are similar but less than that of the deep groundwater. Groundwater at the borehole may circulate at a depth between the above two. The results of this study improve our understanding of the complex groundwater circulation and enable us to better protect and manage the groundwater resources in the region.

**Keywords:** chemical constituents; $^2$H and $^{18}$O; apparent radium age; groundwater flow systems

## 1. Introduction

Groundwater circulation is an important part of the hydrologic cycle. During the circulation, groundwater continuously interacts with geological materials, and its hydrochemical and isotopic compositions change with space and time due to various geochemical processes and isotopic fractionations [1–4]. Understanding the patterns of groundwater circulation is essential for sustainable management of groundwater resources and ecosystems protection [2,3,5]. Hydrogeochemical tracing is proved to be effective and has been widely used to identify different groundwater flow systems [5–8]. In this study, the chemical and environmental isotopic compositions of groundwater and its interactions with geological formations and other water bodies were analyzed in the Xianshui River fault (XF) region to investigate the complex groundwater circulations in the region.

Located in the northeastern Tibet Plateau, China, the XF is one of the most tectonically active intra-continental faults, with a length of 150 km from Kangding to Daofu [9,10] (Figure 1a). The XF region has undergone multiple tectonic movements, namely, the Indosinian movement, Yanshan movement, and Himalayan movement. The XF zone and its adjacent areas form a parallel pattern of mountains, which strike from northwest to southeast (Figure 1a). Earthquakes take place frequently in the region because of the active XF. More than 20 large earthquakes (M > 6.5) have occurred along this fault since 1700 [9]. There are only a few weather and river gauge stations in the region because of the complex topography and tectonic movements, and, thus, long-term hydrologic and/or climatic records are limited.

Most researches in the region were focused on seismic activities; only a few studies of the hydrochemical characteristics focused on water–rock interactions were available in this area [11,12]. None of the studies are about groundwater origin, recharge and circulation issues in the region, although groundwater is an important water source for the local municipality, industry, and agriculture. Furthermore, groundwater circulation in the XF region has implications for hazard assessments because of the well-known series of seismicity in this region. Thus, such a study is urgently needed to be carried out in order to better protect the precious groundwater and for sustainable economic development of the region.

The objective of this paper is to figure out the origin, recharge and evolution of chemical composition of groundwater and the complex groundwater circulation in the XF zone region, which has never been studied before. Water samples from rainfall, river, springs, and wells in the Xialatuo were collected in August and November of 2017. The major ions, the stable isotopies, and four natural radium isotopes were analyzed. The main sources of groundwater recharge, major chemical processes that occurred in groundwater, and different groundwater flow systems are identified based on the hydrochemical characteristics analysis.

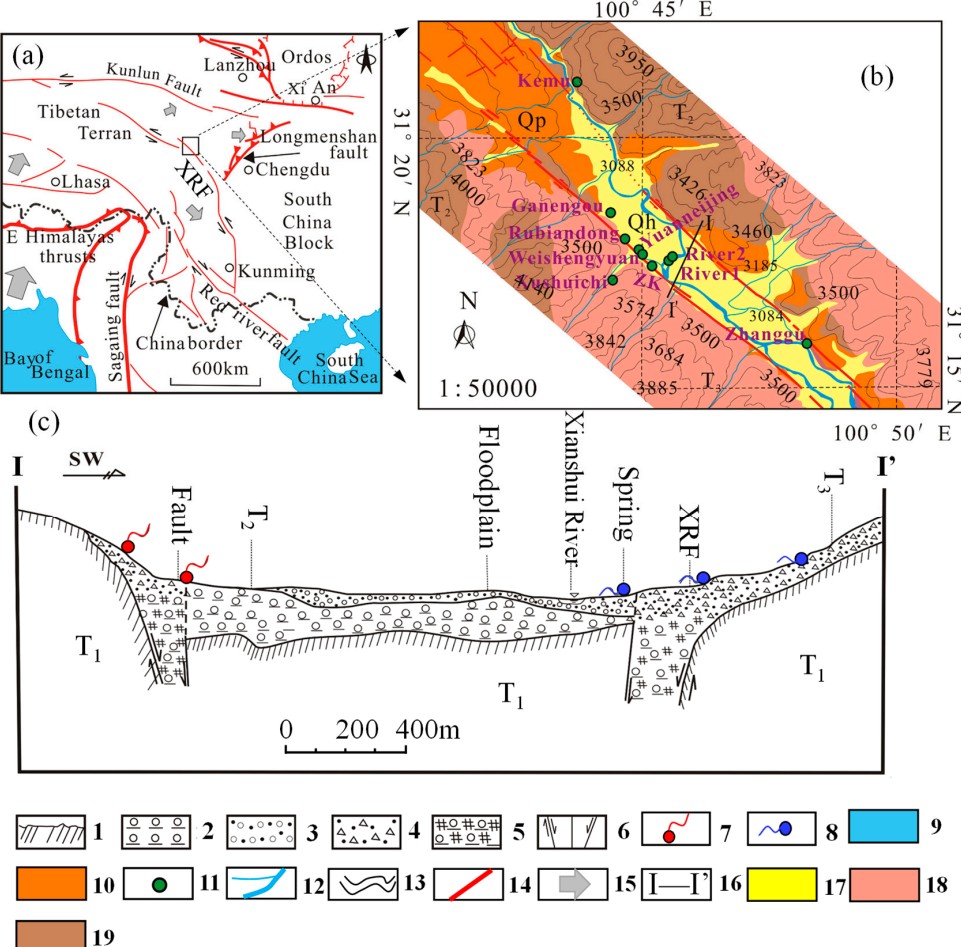

**Figure 1.** (**a**) Location of the Xianshui River fault and the study area (inside the small rectangular area), (**b**) geological map with the sampling points, (**c**) the cross section of I–I' in (**b**) (1. the Upper Triassic; 2. the Middle Triassic; 3. Alluvium of the upper Pleistocene–Holocene; 4. the Lower Triassic; 5. bedrock fractured zone; 6. fault; 7. geothermal spring; 8. cold spring; 9. sea; 10. Qp and 17. Qh represent different layers of the Quaternary; 11. sampling sites; 12. streams; 13. elevations; 14. faults; 15. direction of movements; 16. cross-section; 18. T3; 19. T2.) (Figure 1c was modified after [13]).

## 2. Study Area and Its Hydrogeological Setting

The study area is located to the southeast of the Xialatuo basin in the northwest part of the XF (Figure 1), which is formed by the left-lateral strike-slip movement of XF [9,10]. It belongs to the typical alpine mountain climate zone of the Qinghai–Tibet Plateau. The annual average temperature is 7.4 °C, and the annual total precipitation is 572.5 mm [12]. The Xianshui River, the only one surface water in the study area, flows across the study area from northwest to southeast, and it is recharged by creeks and springs from mountains on both sides (Figure 1b).

The formations in the XF region range from Neoproterozoic to Cenozoic. From Neoproterozoic to Carboniferous, formations are shallow marine carbonates consisting of phyllites, dolomites, marble and evaporates. The lower Permian mainly consists of crystalline and breccia limestone, slate, and phyllites. The upper Permian formations are mainly altered basalt, volcanic agglomerate, breccia limestone, sandstone and dolomites. Triassic covers nearly the entire region with thick deposit. The lower Triassic ($T_3$ in Figure 1) is a deposit with the thickness of about 185~410 m, mainly consisting of slate, siliceous rocks, limestone, and breccia. The middle Triassic ($T_2$ in Figure 1), with a deposit of thickness of about 400~600 m, mainly consists of sandstone and slate. The upper Triassic ($T_1$ in Figure 1), with a deposit of thickness of about 1000~3000 m, mainly consists of sandstone, slate,

and conglomerate. Cretaceous and Jurassic formations are absent due to uplift and denudation in this region. Quaternary sediment almost covers the entire basin [10,12–14].

The main aquifers in the XF region mainly consist of fractured metamorphic rocks of the upper Cambrian series to the upper Permian, sand–slate rocks of Triassic, sandstone and mudstone of Paleogene, and granites formed from different geological periods. Affected by multiple tectonic movements, a series of secondary faults have formed along XF. Two faults developed in the study area (Figure 1b). The Xianshui River fault formed at the south boundary of the basin, strikes NNW–SSE and dips NE at about 85°. The Daofu fault formed at the north boundary of the basin, strikes NNW–SSE and dips NE at about 90°. The direction of groundwater flow in the study area is controlled by topography from the northwest to the southeast. Groundwater discharges through springs and mountain creeks.

There are more than 200 springs [15,16] developed along the XF zone which record important subsurface hydrochemical information through chemical ions and thermal characteristics [11,12]. Zhao [17] indicated that geothermal springs developed along the XF region were controlled by the regional geothermal system through the faults. Five springs (Xushuichi, Rubiandong, Kemu, Zhanggu, Ganengou) occur in the study area. The discharges of Rubiandong and Ganengou spring are less than 1 L/s, while those of Kemu, Zhanggu and Xushuichi spring are much larger than 1 L/s. Xushuichi and Rubiandong located on the hillside are cold springs. Ganengou is a cold spring occurring at the foot of the mountains. Kemu and Zhanggu are hot springs occurring along the Daofu fault.

There are only three wells in the study area during the two sampling campaigns. Yuanneijing drilled many years ago is a domestic well with a depth of 10 m. Weishengyuan built later in 2017 is another domestic well with a depth of 25 m. ZK, drilled in 2015, is a borehole with a depth of 200 m. The borehole ZK (Figure 1b) was drilled at 100°45′7.28″ E, 31°17′34.18″ N with an altitude of 3140 m. Quaternary sediments exposed by drilling contained breccia with mud gravel; the underlying slate rock of Triassic fractures developed well. Most drill cores were short or even broken into pieces along the joint fissures. The hydraulic conductivity of the formation around the borehole ZK was $3.48 \times 10^{-6}$ m/s as obtained from pumping tests.

## 3. Sampling and Analysis Methods

There are no other water sampling sites in the study area besides the Xianshui River, five springs and three wells. The sampling locations are shown in Figure 1b. Therefore, nineteen water samples were collected for chemical and isotopic analysis during 1–3 August 2017 and 15–18 November 2017, including two precipitation samples, three river samples, five well samples and nine spring samples. Temperature, pH and alkalinity of water samples were measured in the field. Temperature and pH were measured using the Hach HQ-40d tester (Hach Company, Lovelan, CO, USA). Alkalinity was measured using a digital titration technique (Hach 1690001) with 0.16 N sulfuric acid (alkalinity < 40 mg/L) and 1.6 N sulfuric acid (alkalinity > 40 mg/L). Two sampling campaigns were conducted to investigate whether season changes had any effect on the physical and chemical properties of different water bodies in the study area.

Water samples for chemical composition and stable isotope analyses were all passed through 0.45 μm filters before being placed in 50 mL polyethylene tubes. All samples were sealed with film immediately and stored in a refrigerator at 4 °C until further analysis. Water samples for $^2H$ and $^{18}O$ isotopic composition analyses were placed in 2 mL glass vials and sealed with film. Radium radioisotopes were collected according to the methods introduced by Moore and Reid [18]. Large-volume (54 L) samples were pumped slowly (less than 2 L/min) through columns containing acrylic fibers impregnated with manganese dioxide (called Mn-fiber), which has been proven to be very effective for extracting radium from groundwater [19]. These Mn-fibers were used to quantitatively sorb the dissolved radium from water, after which they were sent back to the laboratory within 3 days where they were then washed thoroughly to remove all particles for testing.

Major ions (e.g., $Cl^-$, $NO_3^-$, $SO_4^{2-}$, $Ca^{2+}$, $Mg^{2+}$, $Na^+$, and $K^+$) were analyzed with a Dionex ICS-1100 ion chromatography system. Analytical uncertainties were about 2.5%~7% for major anions

and cations. $HCO_3^-$ was measured on the spot using digital titration technique. $CO_3^{2-}$ was not tested here due to the pH of water samples being above 7, the water being alkaline and the content of $CO_3^{2-}$ being too low to detect. All the ion analyses were performed at the Hydrogeological Laboratory of Hong Kong University, China.

STable $^2$H and $^{18}$O isotopic compositions were measured with a Los Gatos IWA-35d Laser Spectrometer (Los Gatos Research, Inc., San Jose, CA, USA) at Hunan Normal University, China. The analytical precision was ±0.2‰ for $^2$H and ±0.05‰ for $^{18}$O. The concentration of $^2$H and $^{18}$O in water samples was measured by infrared laser technology, and the ratio of $^{18}$O/$^{16}$O to $^2$H/$^1$H was obtained by calibration of a standard sample. Isotopic values, the abundance ratio of heavy to light isotope of a sample relative to SMOW (standard mean ocean water), were expressed by the per mil (‰) difference using delta (δ) notation.

$^{223}$Ra, $^{224}$Ra and $^{228}$Ra were tested via a delayed coincidence counting system (RaDeCC System) [20,21] at China University of Geosciences, Beijing. All measurements with RaDeCC System lasted for more than 3 h to minimize the uncertainties, and the uncertainties of $^{223}$Ra, $^{224}$Ra and $^{228}$Ra were 10%, 6% and 7%, respectively [22]. $^{226}$Ra was counted by measuring $^{222}$Rn and its daughters after sealing the fiber in airtight columns to allow for ingrowth. These columns were later mounted to a radon emanation line or a commercially available radon-in-air monitor (RAD 7, Durridge Co.) to measure $^{222}$Rn as a proxy for the $^{226}$Ra [23]. The uncertainty of $^{226}$Ra measured by RAD 7 was 10%.

## 4. Results

### 4.1. Hydrochemical Characteristics

Chemical constituents of the water samples in the two sampling campaigns are presented in Table 1. The charge balance between cations and anions is calculated by $E = \left| \frac{\sum Z\, m_c - \sum Z\, m_a}{\sum Z\, m_c + \sum Z\, m_a} \right| \times 100\%$ [24], where E is the deviation of ions balance, Z is the charge number, $m_c$ and $m_a$ are molar concentrations of cation and anion, respectively. The charge balances ≤ ±5% is within acceptable limits of WHO [25], confirming reliability of the analytical result.

The pH values of the water samples ranged between 7.11 and 7.84 with an average of 7.58. The temperatures of water samples in the first sampling campaign were in the range of 17.6~19.8 °C except for those at Zhanggu (38.6 °C) and Kemu (28.0 °C) hot springs. The temperatures of water samples in the second sampling campaign ranged between 12.0 and 14.5 °C except for those at Zhanggu (32.2 °C) and Kemu (5.5 °C). The total dissolved solids (TDS) (Table 1) of water samples were less than 500 mg/L except for that of Zhanggu.

Hierarchical cluster analysis (HCA) is an effective tool to classify groundwater based on similar hydrogeochemical compositions. As shown in the dendrogram (Figure 2a), a phenon line with a linkage distance of 5 was selected, and three groups of the water samples were divided by the HCA results. The hydrogeochemical processes of the identified groundwater groups can help interpret the relationship between the hydrochemical characterization and flow paths of different groundwater flow systems [5,26]. As shown in Figure 2b, the hydrogeochemical evolution processes presented by the arrowed line indicated that water samples in group I characterized by $HCO_3$-Ca-Mg or $HCO_3$-Mg-Ca type may represent the surface water and shallow groundwater at the recharge and runoff area, while samples in group II characterized by $HCO_3$-Na type may represent the deep groundwater at the discharge area. The samples between the above two groups characterized by $HCO_3$-Na-Mg type may represent the groundwater circulated at a depth between the above two [5,26].

**Table 1.** Results of chemical analyses of the water samples collected during two sampling campaigns (note: 1 represents the first sampling campaign, and 2 represents the second sampling campaign). Unit: mg/L.

| Sites | Note | elev/m | T (°C) | pH | $Na^+$ | $K^+$ | $Mg^{2+}$ | $Ca^{2+}$ | $Cl^-$ | $NO_3^-$ | $SO_4^{2-}$ | $HCO_3^-$ | $F^-$ | TDS | E (%) | Water Type |
|---|---|---|---|---|---|---|---|---|---|---|---|---|---|---|---|---|
| **First Sampling Campaign (2 August 2017)** | | | | | | | | | | | | | | | | |
| Xushuichi1 | spring | 3262 | 18.0 | 7.54 | 8.880 | 0.720 | 42.20 | 45.75 | 1.07 | 0.62 | 11.06 | 330.0 | 0.19 | 275.31 | 3.9 | $HCO_3$-Mg-Ca |
| River1 | river | 3112 | 17.6 | 7.83 | 5.500 | 1.080 | 20.71 | 36.00 | 1.09 | 1.55 | 20.13 | 182.0 | 0.09 | 177.06 | 3.8 | $HCO_3$-Ca-Mg |
| Yuanneijing1 | well | 3148 | 19.8 | 7.44 | 25.99 | 1.65 | 41.40 | 56.56 | 4.09 | 14.36 | 32.78 | 397.0 | 0.23 | 375.33 | −2.5 | $HCO_3$-Mg-Ca |
| Rubiandong1 | spring | 3180 | 17.8 | 7.65 | 40.88 | 3.1 | 50.68 | 40.60 | 2.61 | 2.57 | 16.51 | 423.0 | 0.23 | 368.45 | 3.9 | $HCO_3$-Mg-Ca |
| Kemu1 | spring | 3192 | 28.0 | 7.64 | 7.98 | 0.62 | 18.78 | 41.43 | 0.98 | 1.49 | 9.20 | 250.0 | 0.14 | 205.48 | −4.8 | $HCO_3$-Ca-Mg |
| Zhanggu1 | spring | 3110 | 38.6 | 7.11 | 640.8 | 15.42 | 10.71 | 19.54 | 12.21 | 4.99 | 3.06 | 1650.0 | 2.51 | 1531.7 | 4.1 | $HCO_3$-Na |
| ZK1 | well | 3140 | 14.5 | 7.72 | 33.77 | 3.19 | 5.94 | 7.03 | 5.25 | 0.06 | 9.71 | 132.0 | 0.07 | 128.95 | −1.8 | $HCO_3$-Na-Mg |
| Yushui1 | rain | 2978 | 19.5 | 7.21 | 2.45 | 1.01 | 0.37 | 0.97 | 0.21 | 0.25 | 0.11 | 13.0 | 0.06 | 11.87 | −4.1 | $HCO_3$-Na-Ca |
| Yushui(ce)1 | rain | 3148 | 18.0 | 7.34 | 2.60 | 1.25 | 0.60 | 1.41 | 0.10 | 0.48 | 0.10 | 15.0 | 0.08 | 14.04 | −3.8 | $HCO_3$-Na-Ca |
| **Second Sampling Campaign (16 November 2017)** | | | | | | | | | | | | | | | | |
| Xushuichi2 | spring | 3162 | 14.1 | 7.82 | 10.5 | 0.38 | 38.80 | 39.50 | 1.64 | 0.66 | 14.58 | 308.0 | 0.11 | 260.08 | 1.9 | $HCO_3$-Mg-Ca |
| River2-1 | river | 3112 | 12.0 | 7.94 | 5.68 | 0.62 | 20.87 | 43.38 | 1.61 | 2.64 | 29.66 | 203.0 | 0.13 | 205.96 | 0.8 | $HCO_3$-Ca-Mg |
| River2-2 | river | 3115 | 12.0 | 8.14 | 5.5 | 0.62 | 20.35 | 42.23 | 1.46 | 2.67 | 26.85 | 186.0 | 0.11 | 192.68 | 3.8 | $HCO_3$-Ca-Mg |
| Yuanneijing2 | well | 3148 | 14.4 | 7.64 | 24.10 | 0.59 | 40.58 | 43.00 | 4.03 | 6.66 | 38.48 | 352.0 | 0.16 | 353.44 | −2.7 | $HCO_3$-Mg-Ca |
| Rubiandong2 | spring | 3180 | 14.2 | 7.79 | 43.10 | 0.94 | 51.86 | 34.01 | 2.54 | 2.16 | 15.53 | 415.0 | 0.10 | 357.64 | 3.9 | $HCO_3$-Mg-Ca |
| Kemu2 | spring | 3192 | 5.5 | 7.50 | 11.55 | 0.44 | 28.04 | 49.66 | 1.95 | 2.39 | 14.69 | 270.0 | 0.11 | 243.72 | 4.2 | $HCO_3$-Ca-Mg |
| Zhanggu2 | spring | 3110 | 32.2 | 7.15 | 359.54 | 1.15 | 9.19 | 13.85 | 9.81 | 1.24 | 2.07 | 1123.0 | 3.02 | 958.35 | −5.0 | $HCO_3$-Na |
| ZK2 | well | 3140 | 13.1 | 7.80 | 30.01 | 4.29 | 5.61 | 8.19 | 6.66 | 0.11 | 6.95 | 114.0 | 0.03 | 115.82 | 1.2 | $HCO_3$-Na-Mg |
| Weishengyuan | well | 3151 | 13.5 | 7.84 | 18.25 | 0.98 | 36.75 | 41.84 | 11.55 | 26.42 | 25.34 | 290.0 | 0.09 | 306.13 | −4.3 | $HCO_3$-Mg-Ca |
| Ganengou | spring | 3166 | 14.5 | 7.50 | 11.55 | 0.61 | 33.02 | 61.68 | 9.50 | 5.90 | 10.59 | 327.0 | 0.20 | 296.35 | 2.1 | $HCO_3$-Ca-Mg |

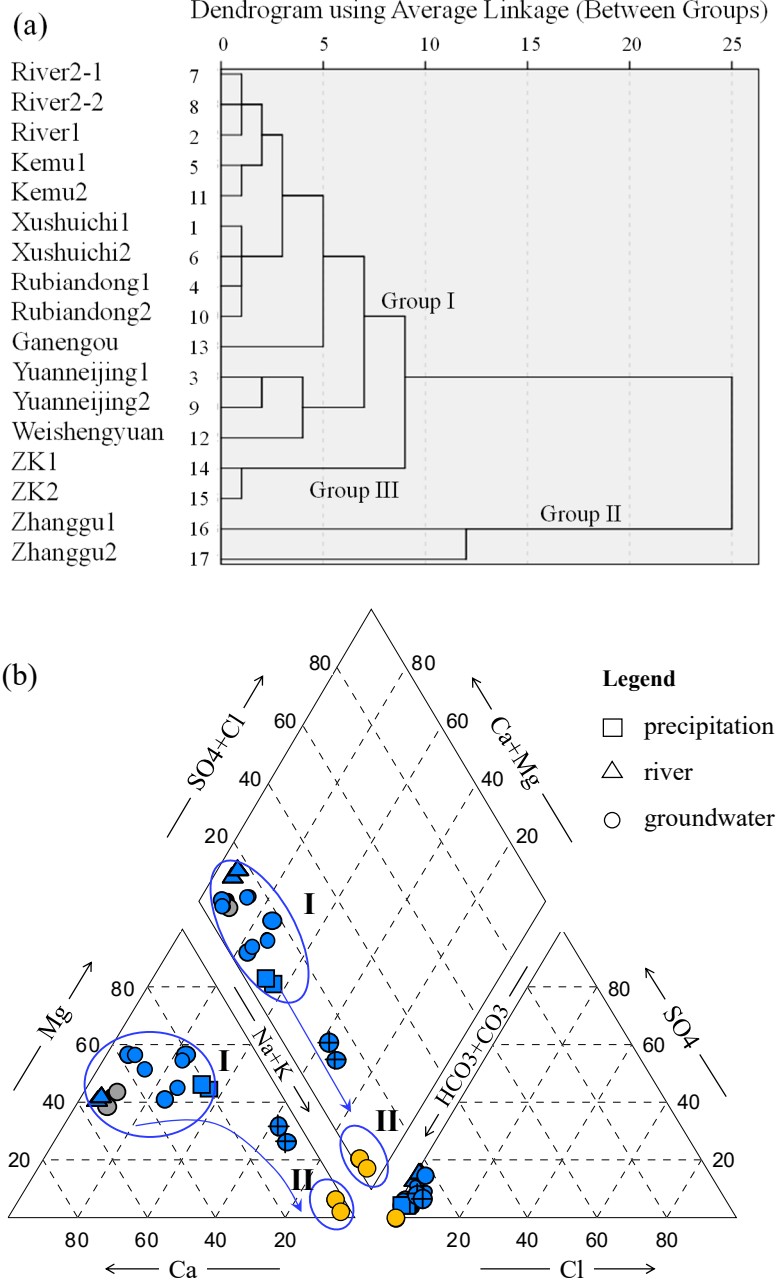

**Figure 2.** (**a**) Dendrogram showing the three groups of the water samples; (**b**) Piper plot of all 19 water samples collected in two sampling campaigns. Squares are rain samples, triangles are river samples, circles are groundwater samples, and circles with a cross are the water samples at the borehole ZK. Yellow represents the spring at Zhanggu, grey represents the spring Kemu, and blue represents the rest.

Minor elements (e.g., F ion in this study) can provide additional information on groundwater evolution because the concentrations of minor constituents are usually controlled by water–rock reactions [11]. A linear relationship between F$^-$ and temperature in the water samples (Figure 3) excluding Zhanggu (its extremely high F ion contents and temperature resulting in an unclear comparison among the water samples shown in the small figure in Figure 3) was displayed, indicating that the groundwater circulation spans different depths [27,28]. The high F$^-$ and high temperature (Table 1) in Zhanggu indicated that Zhanggu spring circled deeper than other water samples. Kemu spring, like Zhanggu, is also a geothermal spring. However, F$^-$ contents of Kemu are much lower than that of Zhanggu, inferring that Kemu spring was probably recharged a lot by sources

with low F⁻, like precipitations, shallow ground or surface waters through weathered fissure and/or tectonic fractures. This point was also evidenced by the water temperature at Kemu, which is 28.0 °C higher than the local air temperature (23.4 °C) during the first sampling campaign (summer) and is 5.5 °C lower than the local air temperature (7.1 °C) during the second sampling campaign (winter).

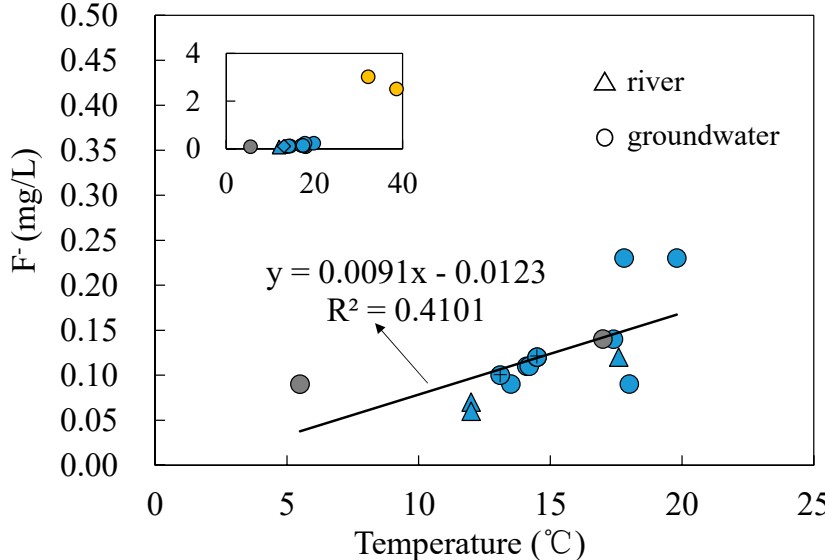

**Figure 3.** Relationship between F ion and temperature excluding the hot spring at Zhanggu. Triangles are river samples, circles are groundwater samples, and circles with a cross are the water samples at the borehole ZK. Yellow represents the spring at Zhanggu, grey color represents the spring at Kemu, and blue represents the rest.

*4.2. Stable Isotopic Compositions*

The $\delta^2 H$ and $\delta^{18} O$ values of the water samples in the two sampling campaigns are shown in Table 2. With regard to the standard mean ocean water (SMOW), stable isotopic compositions of water samples varied from −135.56 to −100.68‰ for $\delta^2 H$ and from −17.58 to −13.96‰ for $\delta^{18} O$. The local meteoric water line (LMWL) was developed by Bershaw [29] as $\delta^2 H = 7.67 \delta^{18} O + 6.7$. The smaller slope of the LMWL compared to the global meteoric water line (GMWL, $\delta^2 H = 8 \delta^{18} O + 10$) revealed the isotopic composition characteristic of atmospheric precipitation in the study area where secondary evaporation during rainfall was expected [5,30]. With the reference of the LMWL, all water samples distributed along the LMWL, precipitation origin was suggested for different waters in the study area.

The deuterium shift is defined as ($\delta^2 H_m - \delta^2 H_{th}$), where $\delta^2 H_m$ is the measured deuterium value of the water samples and $\delta^2 H_{th}$ is the theoretical deuterium value according to the LMWL. The deuterium shifts of Rubiandong, Kemu, Zhanggu samples in the two sampling campaigns were approximately 1.67‰, 0.78‰, and 2.0‰, which should correspond to 0.21‰, 0.10‰, and 0.25‰ in the shift of $\delta^{18} O$. However, the very different changes (0.04‰, 0.66‰, 0.54‰) of the $\delta^{18} O$ shift were observed in the data. These different shifts in the $\delta^{18} O$ content of water were attributed to isotopic exchange between the host rock and water [27,30]. In such a groundwater flow system, the range in $\delta^2 H$ content reflects the original composition of input water undergoing deep circulation and the impact of shallower more rapid recharge water mixing as the water ascends [11,27,31].

**Table 2.** The $^2$H and $^{18}$O activities of the water samples in two sampling campaigns (note: 1 represents the first sampling, and 2 represents the second sampling). Unit: ‰.

| First Sampling Campaign (2 August 2017) | | | | |
|---|---|---|---|---|
| **Sites** | $\delta^2$**H** | **StDev($\delta^2$H)** | $\delta^{18}$**O** | **StDev($\delta^{18}$O)** |
| Yushui1 | −100.68 | 0.189 | −14.14 | 0.056 |
| Yushui (ce)1 | −110.72 | 0.602 | −13.96 | 0.114 |
| River1 | −114.18 | 0.300 | −15.48 | 0.099 |
| Yuanneijing1 | −114.19 | 0.356 | −14.99 | 0.055 |
| Xushuichi1 | −119.88 | 0.484 | −15.65 | 0.077 |
| ZK1 | −126.60 | 0.485 | −16.82 | 0.055 |
| Rubiandong1 | −127.61 | 0.216 | −16.12 | 0.097 |
| Kemu1 | −126.83 | 0.085 | −16.79 | 0.063 |
| Zhanggu1 | −133.56 | 0.363 | −17.04 | 0.131 |
| Second Sampling Campaign (16 November 2017) | | | | |
| **Sites** | $\delta^2$**H** | **StDev($\delta^2$H)** | $\delta^{18}$**O** | **StDev($\delta^{18}$O)** |
| River2-1 | −109.32 | 0.310 | −15.06 | 0.054 |
| River2-2 | −110.06 | 0.204 | −15.12 | 0.111 |
| Yuanneijing2 | −120.69 | 0.521 | −15.86 | 0.049 |
| Xushuichi2 | −117.27 | 0.608 | −15.53 | 0.089 |
| ZK2 | −126.16 | 0.168 | −16.98 | 0.044 |
| Rubiandong2 | −125.94 | 0.611 | −16.16 | 0.070 |
| Kemu2 | −127.60 | 0.238 | −17.45 | 0.073 |
| Zhanggu2 | −135.56 | 0.345 | −17.58 | 0.048 |
| Weishengyuan | −118.11 | 0.147 | −15.77 | 0.089 |
| Ganengou | −116.85 | 0.199 | −15.22 | 0.074 |

### 4.3. Radium Isotopes

The activity concentrations of four natural radium isotopes of the water samples in the two sampling campaigns are presented in Table 3. Activity concentrations of $^{224}$Ra ranged from 10.5 to 70.8 dpm/100 L in the first sampling and from 12.2 to 86.6 dpm/100 L in the second sampling, respectively (Table 3). Activity concentrations of $^{223}$Ra ranged from 0.05 to 0.68 dpm/100 L in the first sampling and from 0.06 to 0.66 dpm/100 L in the second sampling (Table 3). The activity concentrations of $^{226}$Ra ranged from 3.2 to 8.24 dpm/100L in the first sampling and from 1.59 to 10.96 dpm/100L in the second sampling. The activity concentrations of $^{228}$Ra ranged from 1.81 to 21.04 dpm/100 L in the first sampling and from 2.48~29.69 dpm/100 L in the second sampling. The radium activities of the water samples were within the radium activity range of groundwater documented in previous studies [32,33], except for those of Zhanggu spring in which the concentrations of $^{223}$Ra, $^{224}$Ra, $^{226}$Ra and $^{228}$Ra were 33.10, 531.22, 411.02 and 12,740.96 dpm/100 L, respectively, in the first sampling campaign and 33.05, 367.62, 355.02 and 18,365.4 dpm/100 L, respectively, in the second sampling campaign.

The relationship between the four natural radium isotopes and TDS (excluding Zhanggu because of its extremely high radium resulting in an unclear comparison among the water samples) is shown in Figure 4a. It is well known that TDS is an important factor that determines the concentration of Ra isotopes [34–36], however, the correlation between TDS and $^{223}$Ra ($R^2 = 0.317$, *p*-value = 0.02), $^{224}$Ra ($R^2 = 0.39$, *p*-value = 0.01), $^{226}$Ra ($R^2 = 0.097$, *p*-value = 0.25), $^{228}$Ra ($R^2 = 0.35$, *p*-value = 0.02) in this study is insignificant, which may be attributed to the well-mixed processes of the groundwater during the circulation in the study area. The extremely high activity concentrations of $^{224}$Ra and $^{223}$Ra ($t^{1/2}$ of $^{224}$Ra = 3.6 d, $t^{1/2}$ of $^{223}$Ra = 11.4 d) in the groundwater at Zhanggu may be due to the fact that the circulation of the groundwater moved quickly through the developed fractures of XF.

**Table 3.** Activity concentrations and activity ratios of the radium isotopes in the water samples (note: 1 represents the first sampling, and 2 represents the second sampling). Unit: dpm/100 L.

| Sites | $^{223}$Ra(error) | $^{224}$Ra(error) | $^{226}$Ra(error) | $^{228}$Ra(error) | $^{224}$Ra/$^{223}$Ra | $^{224}$Ra/$^{226}$Ra |
|---|---|---|---|---|---|---|
| **First Sampling Campaign (2 August 2017)** | | | | | | |
| Xushuichi1 | 0.14(0.02) | 70.0(3.5) | 4.43(0.2) | 15.44(1.7) | 500.00 | 15.80 |
| River1 | 0.05(0.01) | 70.8(3.6) | 3.20(0.4) | 1.81(0.3) | 1416.00 | 22.13 |
| Yuanneijing1 | 0.45(0.05) | 45.3(2.3) | 4.16(0.5) | 19.00(1.5) | 100.67 | 10.89 |
| Rubiandong1 | 0.68(0.07) | 53.9(2.7) | 8.24(1.2) | 21.04(1.4) | 79.26 | 6.54 |
| Kemu1 | 0.15(0.02) | 50.2(2.51) | 3.74(0.5) | 8.48(1.3) | 333.33 | 13.37 |
| ZK1 | 0.18(0.02) | 10.5(0.53) | 7.61(1.1) | 14.51(1.1) | 55.56 | 1.31 |
| Zhanggu1 | 33.10(3.30) | 531.22(26.56) | 411.02(28.5) | 12,740.96(891.9) | 16.05 | 1.29 |
| Mean | 5.06 | 85.04 | 62.73 | 2017.47 | 357.27 | 10.19 |
| **Second Sampling Campaign (16 November 2017)** | | | | | | |
| Xushuichi2 | 0.21(0.02) | 62.4(3.1) | 3.24(0.2) | 18.91(1.2) | 297.14 | 19.26 |
| River2-1 | 0.07(0.01) | 56.0(2.8) | 1.93(0.1) | 2.48(1.4) | 800.00 | 29.02 |
| Yuanneijing2 | 0.36(0.04) | 103.1(5.2) | 8.77(0.3) | 29.69(1.3) | 286.39 | 11.76 |
| Rubiandong2 | 0.51(0.05) | 86.6(4.3) | 6.36(0.4) | 29.20(1.1) | 169.80 | 13.62 |
| Kemu2 | 0.66(0.07) | 72.6(3.6) | 9.43(0.5) | 19.94(1.5) | 110.00 | 7.70 |
| ZK2 | 0.38(0.02) | 12.2(0.6) | 6.61(0.4) | 20.32(1.1) | 31.58 | 1.82 |
| Zhanggu2 | 33.05(3.31) | 367.62(18.4) | 355.02(10.85) | 18,365.4(1285.6) | 11.12 | 1.04 |
| River2-2 | 0.06(0.01) | 47.0(2.4) | 1.59(0.11) | 3.18(0.8) | 783.33 | 29.56 |
| Weishengyuan | 0.28(0.03) | 79.8(4.0) | 10.96(0.42) | 15.92(1.6) | 285.71 | 7.30 |
| Ganengou | 0.30(0.02) | 60.6(3.0) | 8.04(0.56) | 27.20(1.2) | 200.00 | 7.50 |
| Mean | 3.59 | 94.73 | 41.19 | 200.21 | 297.51 | 12.86 |

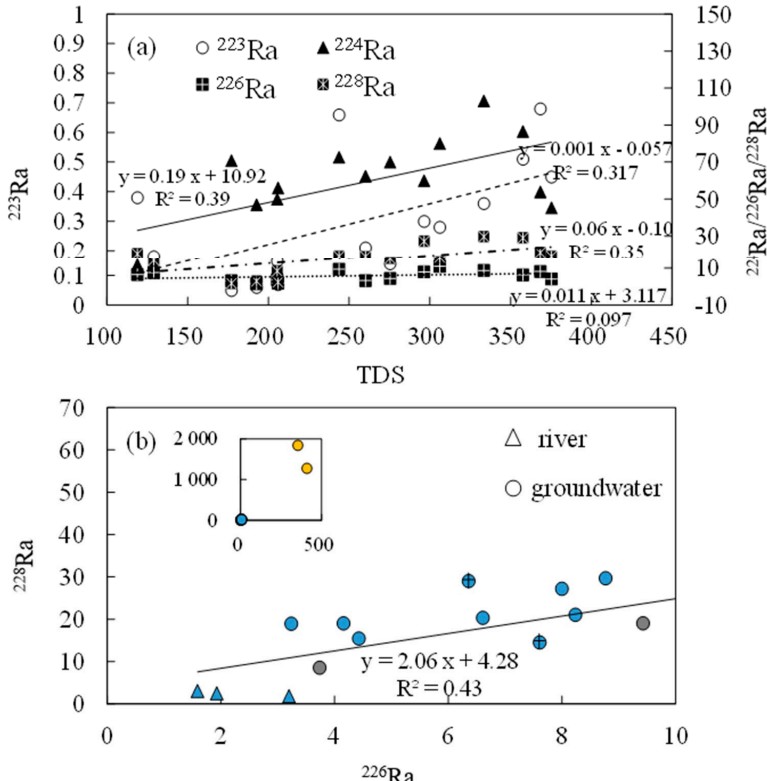

**Figure 4.** (**a**) Relationship between the activity concentrations of four natural radium isotopes and TDS in the water samples, excluding the hot spring at Zhanggu which contains extremely high salinity and radium contents. (**b**) Activity concentration of $^{228}$Ra as a function of the activity concentration of $^{226}$Ra. Triangles are river samples, circles are groundwater samples, and circles with a cross are the water samples at the borehole ZK. Yellow represents the spring at Zhanggu, grey represents the spring at Kemu, and blue represents the rest.

$^{226}$Ra (t$^{1/2}$ = 1600 a) is the daughter of $^{230}$Th in the $^{238}$U decay series. $^{228}$Ra (t$^{1/2}$ = 5.75 a) is the daughter of $^{232}$Th. The activity concentration of $^{228}$Ra is a function of the activity concentration of $^{226}$Ra in the groundwater which is dependent on the ratio of Th/U (Th/U = 2.7 of 41 rock samples in the study area) in the host rock [32,34]. Although the correlations between $^{228}$Ra and $^{226}$Ra were positive (Figure 4b), a large scattering occurred, excluding Zhanggu, which may be attributed to the water mixing processes that occurred during the groundwater circulation [34,35,37].

## 5. Discussion

### 5.1. Isotopic Characteristics Associated with Different Groups

The hydrochemical and isotopic compositions of groundwater change through continuously various geochemical processes and isotopic fractionation along the flow path during circulation. The stable hydrogen and oxygen isotopes have been proven to be a special and powerful tool to trace groundwater origin and circulations within different flow systems [4,5,8,38–40].

The linear regression of the stable hydrogen and oxygen isotopic composition in water samples may indicate close hydraulic connections among groundwater at different depths [5]. The river samples (the triangles shown in Figure 5a) located on the upper right-hand side exhibited a local evaporation line as $\delta^2$H = 11.649$\delta^{18}$O + 66.098. The intercept of the river samples line and the LMWL stood for the original isotopic compositions of the recharged rainfall before evaporation [1], which was −107.8 ‰ for $\delta^2$H and −14.93 for $\delta^{18}$O. The river samples line also intersected with the groundwater line ($\delta^2$H = 7.02$\delta^{18}$O − 9.75) (Figure 5a) at −124.75‰ for $\delta^2$H and −16.38‰ for $\delta^{18}$O, which was between precipitation and deep groundwater (<−120 for $\delta^2$H and <−16 for $\delta^{18}$O in this study), indicating the recharge of the Xianshui River by groundwater and precipitation [5,41]. The Xianshui River is a perennial river located in the discharge zones of groundwater flow systems in this study area, and can received recharge from precipitation and groundwater systems [41].

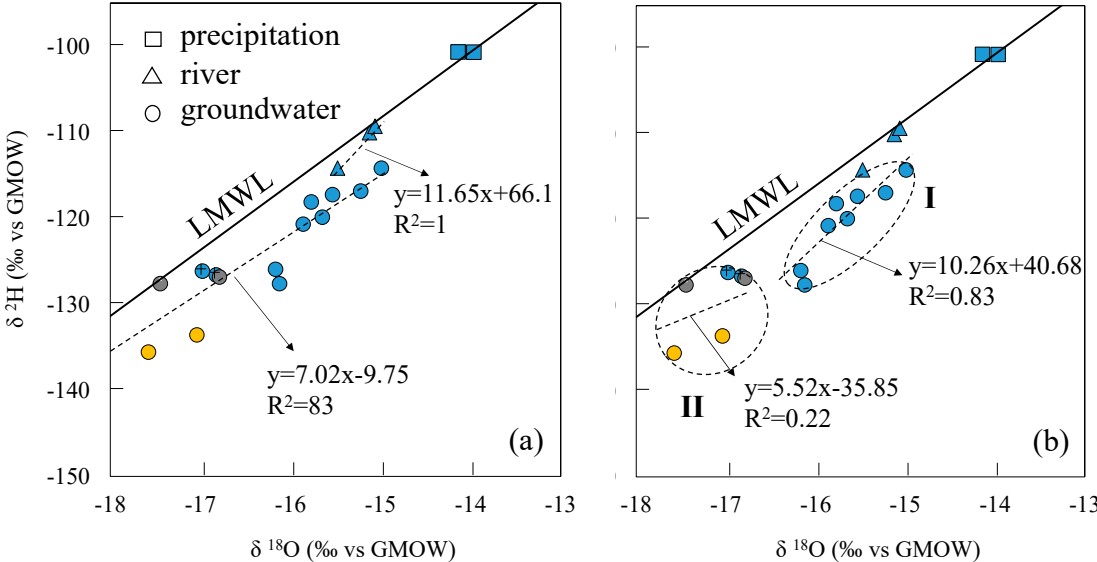

**Figure 5.** (**a**) Relationship between $^2$H and $^{18}$O for the river water and the groundwater samples; (**b**) relationship of $^2$H and $^{18}$O for different groups of the analyzed water samples. LMWL represents the local meteoric water line. Squares are rain samples, triangles are river samples, circles are groundwater samples, and circle with a cross are the water samples at the borehole ZK. Yellow represents the spring at Zhanggu, grey represents the spring at Kemu, and blue represents the rest. Group I represents the shallow groundwater, and group II represents the deep groundwater.

The groundwater samples in group I (Figure 5b) were less depleted isotopically than other groundwater samples with $\delta^2$H > −120‰ and $\delta^{18}$O > −16‰, exhibiting a linear regression line

(the group I line) of $\delta^2H = 10.26\delta^{18}O + 40.68$. Moreover, the intersection of the group I line with the LMWL was −93.93‰ for $\delta^2H$ and −13.12‰ for $\delta^{18}O$, which were close to the average isotopic compositions of precipitation samples ($\delta^2H = -100.70$‰ and $\delta^{18}O = -14.05$‰). Therefore, water samples in group I may be the groundwater circulated at a relative shallow depth [5]. The four hot spring water samples (two samples of Zhanggu and two samples of Kemu) in group II (Figure 5b) were more depleted isotopically than other groundwater samples with $\delta^2H < -130$‰ and $\delta^{18}O < -17$‰ for Zhanggu, and $\delta^2H < -120$‰ and $\delta^{18}O < -16$‰ for Kemu, respectively. The intercept of the hot spring samples line ($\delta^2H = 5.52\delta^{18}O - 35.85$, $R^2 = 0.22$ is caused by the isotopic compositions of Kemu which is a hot spring controlled by the regional geothermal system and probably recharged a lot by precipitations or surface waters) and the LMWL also stood for the original isotopic compositions of the recharged rainfall before evaporation [1], which was −145.09‰ for $\delta^2H$ and −19.79‰ for $\delta^{18}O$, reflecting that the recharge of the hot springs may be the relatively older precipitations. Therefore water samples in group II may be the groundwater circulated at a relatively deep depth. The $\delta^2H$ and $\delta^{18}O$ values of ZK (the borehole with a depth of 200m) were between the shallow groundwater and the deep groundwater, confirming that the groundwater at ZK may be circulated at a depth between the above two.

It is noted that Kemu spring belongs to group II. The isotopic compositions of Kemu spring are much lower than those of Zhanggu spring and much higher than those of water samples in group I, similar to those of water samples in ZK borehole, inferring that the circulation depth of water samples in Kemu is much deeper than those of group I. Geothermal springs in the Xianshui River region are controlled by the regional geothermal system through the faults [17]. As mentioned above, Kemu spring has high water temperature in the summer and low water temperature in the winter, suggesting that Kemu spring is also controlled by the regional geothermal system through the faults and that the water circulation in Kemu is deep, and it is not certain that the circulation depth is the same as that of Zhanggu. Besides, the F- content of Kemu spring is much lower than that of Zhanggu spring. These physical and geochemical properties of water samples in Kemu infer that Kemu spring was probably recharged by precipitations, shallow ground or surface waters through weathered fissure and/or tectonic fractures.

*5.2. Hydrogeochemical Characteristics Associated with Different Groups*

The processes controlling groundwater chemical characteristics in the study area were discussed from the perspective of the leaching effect and ion exchange.

The leaching effect is a process that transfers rock minerals to groundwater during the water–rock interaction, which can be identified by chemical constituents and constituent ratios [3,4,6,11,26,42]. As presented in Figure 6a, the Na/Cl ratios of the water samples were located above the 1:1 dissolution line, indicating that the salinity originated from not only the dissolution of halite [2,3,26] but also the dissolution of silicate minerals or cation exchange [5,6].

The diagram of $Ca^{2+}$ versus $SO_4^{2-}$ with the gypsum dissolution line argues for the role of the dissolution of gypsum as a major process contributing to the groundwater mineralization [2–4,42,43]. The shift of data points from the gypsum dissolution line indicates that gypsum dissolution in groundwater may be coupled by several other processes: (i) cation exchange; (ii) sulfate reduction under anaerobic conditions; (iii) simultaneous incongruent dissolution of dolomite; and (iv) precipitation of carbonates controlled by gypsum dissolution [2,3,42]. As shown in Figure 6b, all water samples were plotted above the gypsum dissolution line, suggesting that the dissolution of gypsum was not the sole source of $Ca^{2+}$, and $Ca^{2+}$ was also influenced by other processes such as cation exchange or simultaneous incongruent dissolution of dolomite and gypsum or precipitation of carbonates controlled by gypsum dissolution [3,42].

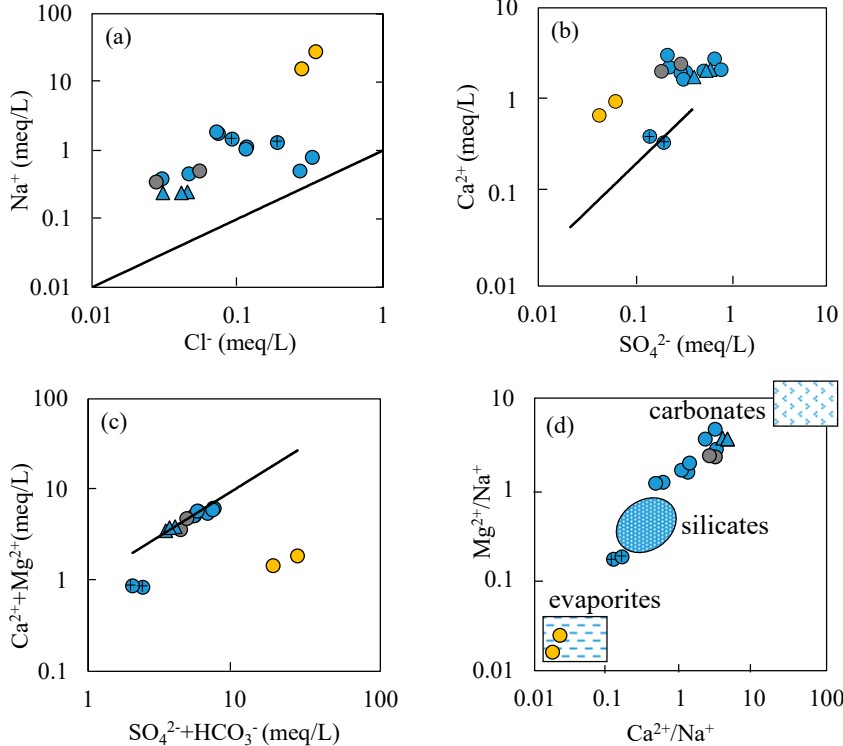

**Figure 6.** Relationship between (**a**) $Na^+$ and $Cl^-$, (**b**) $SO_4^{2-}$ and $Ca^{2+}$, (**c**) $[Ca^{2+} + Mg^{2+}]$ and $[SO_4^{2-} + HCO_3^-]$, and (**d**) $[Ca^{2+}/Na^+]$ and $[Mg^{2+}/Na^+]$. Triangles are river samples, circles are groundwater samples, and circles with a cross are the water samples at the borehole ZK. Yellow represents the spring at Zhanggu, grey represents the spring at Kemu, and blue represents the rest.

Water samples in the Ca + Mg versus $HCO_3$ + $SO_4$ plot (Figure 6c) fell on the 1:1 dissolution line, except for Zhanggu and ZK which were plotted below the 1:1 line, indicating that the chemical constituents of most water samples depended on the dissolution of carbonates, sulfates and silicates. With respect to Zhanggu and ZK, the dissolution of sulfate and silicate minerals contributed most to groundwater chemical constituents [5,26]. The plot of Na-normalized Ca versus Mg (Figure 6d) further indicated the results of Figure 6c. The points data (Figure 6d), excluding those of Zhanggu and ZK, were distributed in the area between the silicates and the carbonates, suggesting that most of the analyzed water samples were mainly influenced by the dissolution of silicates and carbonates; Zhanggu and ZK samples were distributed in the area between the evaporites and the silicates, meaning that the chemical constituents of the groundwater at these sites are likely controlled by the dissolution of the evaporates and silicates [5].

The plot of $[(Ca + Mg) - (HCO_3 + SO_4)]$ versus $[(Na + K) - Cl]$ is widely used to illustrate the effect of ion exchange on the water chemical constituents [2,3,5,11]. As shown in Figure 7a, the points data were plotted near the 1:1 line, showing that the cation exchange was a noteworthy composition-controlling process in the groundwater. The chloro-alkaline indices calculated by chloric alkali index (CAI)-1 (CAI-1 $= \frac{Cl-Na-K}{Cl}$) and CAI-2 (CAI-2 $= \frac{Cl-Na-K}{HCO_3+SO_4+CO_3+NO_3}$) [44] were also indicated in the cation exchange; all ions in the formula are expressed in meq/L. Negative CAI-1 and CAI-2 indicate the exchange of $Ca^{2+}$ and/or $Mg^{2+}$ in groundwater and $Na^+$ and/or $K^+$ in stratum, while positive CAI-1 and CAI-2 indicate the exchange of $Na^+$ and/or $K^+$ in groundwater and $Ca^{2+}$ and/or $Mg^{2+}$ in stratum. As shown in Figure 7b, the CAI-1 and CAI-2 values of the analyzed water samples were negative. The larger the absolute value of CAI-1, the higher the degree of ion exchange [5]. Thus Zhanggu, ZK and Rubiandong samples had a higher degree of ion exchange than other water samples.

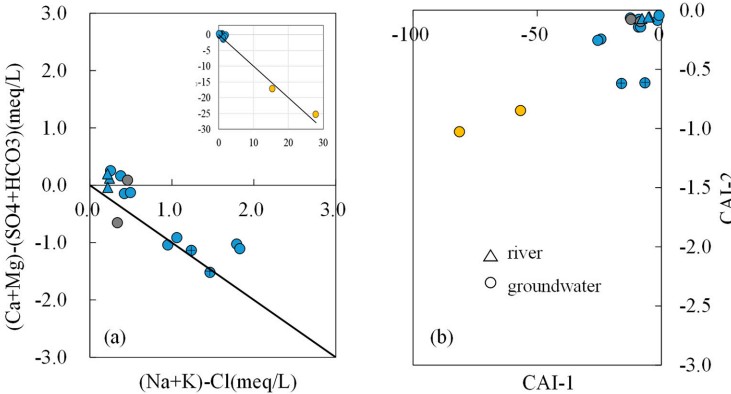

**Figure 7.** Relationships between (**a**) (Na + K) − Cl and (Ca + Mg) − (HCO$_3$ + SO$_4$); (**b**) CAI-1 and CAI-2 (CAI represents chloric alkali index). Triangles are river samples, circles are groundwater samples, and circles with a cross are the water samples at the borehole ZK. Yellow represents the spring at Zhanggu, grey represents the spring at Kemu, and blue represents the rest.

The saturation states (shown in Figure 8) of the water samples with respect to halite, calcite, dolomite and gypsum were computed by the PHREEQC program to better investigate the minerals dissolved or precipitated in the groundwater system [2,3,11,43]. All water samples were under-saturated (SI < 0, SI is saturation index) with respect to halite and gypsum (Figure 8a,d), indicating that evaporates, silicates and sulfates were probably dissolved by water along the flow path. The SI values of most samples (excluding ZK, Zhanggu and Kemu) were greater than zero with respect to calcite and dolomite (Figure 8b,c), indicating that the groundwater with HCO$_3$-Mg-Ca or HCO$_3$-Ca-Mg type was easily oversaturated [5,45]. ZK with HCO$_3$-Na-Mg type was undersaturated (SI < 0) with respect to calcite and dolomite, indicating that the stratum of this site probably had little calcite and dolomite dissolved. The values of SI calculated for the samples in Zhanggu and Kemu were approximately equal to zero, indicating that the groundwater was probably saturated with calcite and dolomite.

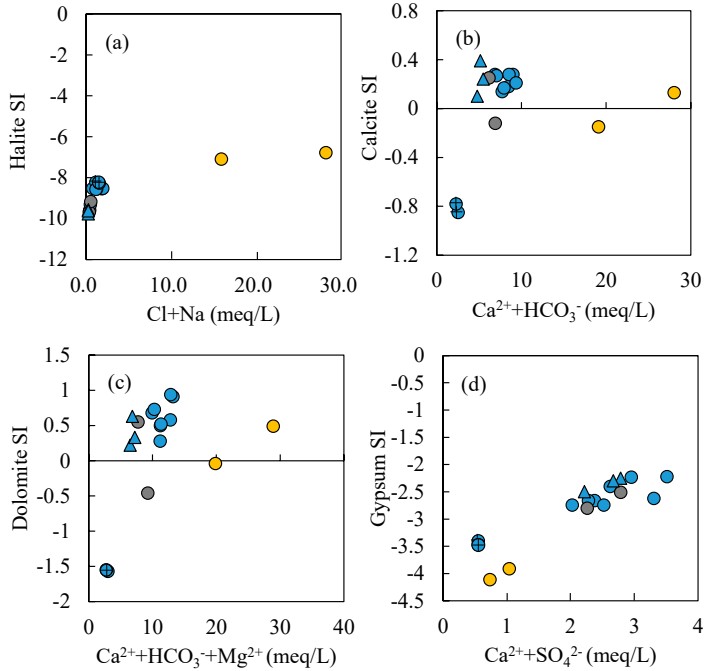

**Figure 8.** Relationship between (**a**) halite SI and Cl + Na, (**b**) calcite SI and Ca + HCO$_3$, (**c**) dolomite SI and Ca + HCO$_3$ + Mg, and (**d**) gypsum SI and Ca + SO$_4$. Triangles are river samples, circles are groundwater samples, and circles with a cross are the water samples at the borehole ZK. Yellow c represents the spring at Zhanggu, grey represents the spring at Kemu, and blue represents the rest.

### 5.3. Characteristics of the Radium Isotopes during the Circulation of Groundwater

The activity concentration of radium in groundwater is controlled by chemical and physical processes such as dissolution, desorption or adsorption from aquifer surfaces; complexation with other adsorbed species; and co-precipitation in minerals [46,47]. Therefore radium concentration or radium ratio exhibits a closer relationship with the groundwater chemistry as well as with the host rock [35,36,47]. As shown in Figure 9a,b, the correlation between $^{228}$Ra and Ca ($R^2 = 0.01$, *p*-value = 0.74), Mg ($R^2 = 0.29$, *p*-value = 0.04), Na ($R^2 = 0.39$, *p*-value = 0.01), and Cl ($R^2 = 0.18$, *p*-value = 0.14), SO$_4$ ($R^2 = 0.16$, *p*-value = 0.11), and HCO$_3$ ($R^2 = 0.37$, *p*-value = 0.01) in the water samples excluding Zhanggu (the abnormally high concentration of radium resulting in an unclear comparison), the correlation between $^{228}$Ra and these ions is not so clear, and there was a large scattering between the data of $^{228}$Ra and these ions in the water samples, indicating that chemical constituents of these water samples may be originated from a well-mixed reservoir [34,35,47].

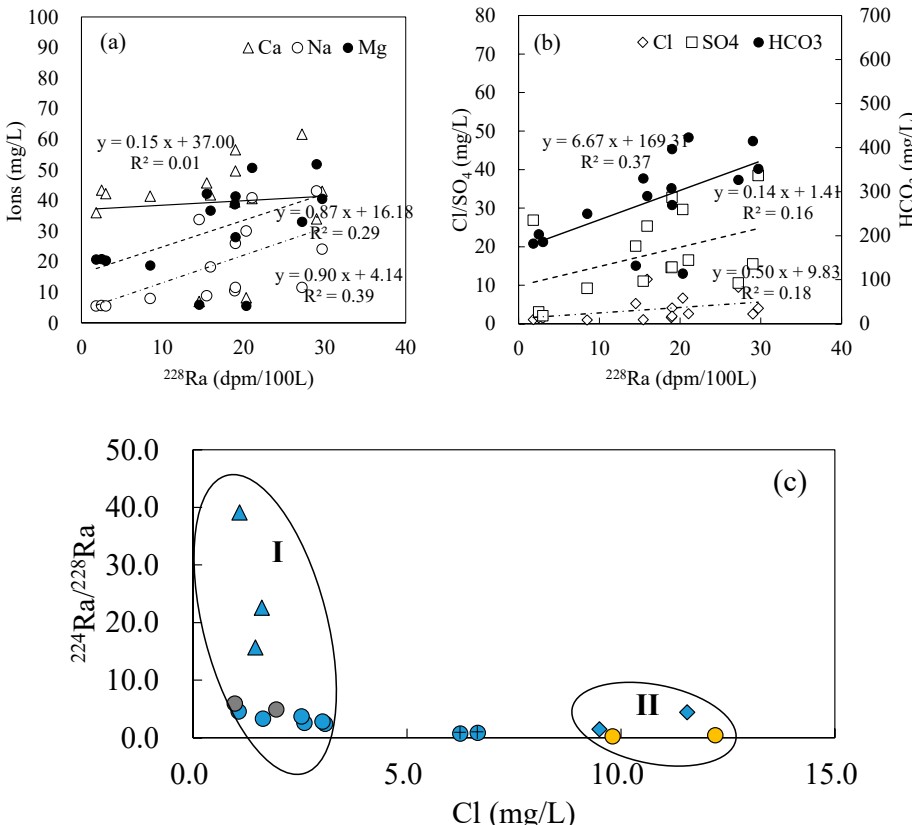

**Figure 9.** Relationship between (**a**) Na, Ca and $^{228}$Ra, (**b**) Cl, SO$_4$ and $^{228}$Ra, and (**c**) $^{224}$Ra/$^{228}$Ra and Cl. Triangles are river samples, circles are groundwater samples, and circles with a cross are the water samples at the borehole ZK. Yellow represents the spring at Zhanggu, grey represents the spring Kemu, diamonds are samples at Weishengyuan and Ganengou, and blue represents the rest.

Chloride (Cl) is considered as a typical tracer because of its relatively conservative chemical property, and it does not easily form minerals or become adsorbed by colloform deposits [11,48]. The ratios of short-lived to long-lived radium may indicate the rates of radium contribution or removal [36], and the lower ratios indicate the radium source from primary mineral dissolution in which short-lived radium decays faster than the dissolution rate, inferring that the circulation flow path is longer. $^{224}$Ra/$^{228}$Ra decreases along the flow paths [32]. The relationship between $^{224}$Ra/$^{228}$Ra ratios and Cl ion is shown in Figure 9c, and the analyzed water samples can be clearly clustered into three groups [34]: Zhanggu samples with low ratios of $^{224}$Ra/$^{228}$Ra and high Cl contents were clustered into group II, inferring that Zhanggu may be circulated at a deeper depth. It is noted that

Weishengyuan (well) and Ganengou (cold spring) (shown by two diamonds) were also clustered into group II because of the high Cl ion contents influenced by significant return flows from human and livestock waste ($NO_3^-$ are more than 10 mg/L in these two sites) (Weishengyuan located in the clinic, Ganengou situated near a livestock shed). The remaining water samples, such as Yuanneijing, Xushuichi and Rubiandong, were clustered into group I with relatively high ratios of $^{224}Ra/^{228}Ra$ and low Cl ion contents, inferring that the groundwater in these sites may be circulated at a relatively shallow depth. Kemu was also clustered into group I, which may be due to it being greatly recharged by infiltrations from precipitations or shallow groundwater through developed fractures, which was also verified by the low TDS and the varied temperatures of the water mentioned above. The circulation depth of groundwater at ZK may be between the above two (Figure 9c).

The "apparent radium age" models [49] using the radium concentration ratio (AR) were applied to determine the apparent radium age of water masses [50,51], which can be described as:

$$t = \ln\left(\frac{AR_{obs}}{AR_i}\right) \times \frac{1}{\lambda_{223} - \lambda_{224}} \tag{1}$$

$$t = \ln\left(\frac{AR_{obs}}{AR_i}\right) \times \frac{1}{\lambda_{226} - \lambda_{224}} \tag{2}$$

where t is the apparent radium age, representing the time that has elapsed since the water sample became enriched in Ra and was isolated from the source. $\lambda_{223} = 0.0608$/day and $\lambda_{224} = 0.189$/day are the decay constants of $^{223}Ra$ and $^{224}Ra$, and the decay of $^{226}Ra$ is neglected because of $\lambda_{226} = 1.18 \times 10^{-6}$/day. $AR_{obs}$ is the activity ratio, $AR_i$ is the initial activity ratio. Using the $AR_{obs}$ measured on the river samples as the $AR_i$ sets the average age of the river waters to zero.

On the basis of both the $^{224}Ra/^{223}Ra$ AR ($AR_i = 1416.0$ of the first sampling campaign, $AR_i = 800.0$ of the second sampling campaign) and the $^{224}Ra/^{226}Ra$ AR ($AR_i = 22.13$ of the first sampling campaign, $AR_i = 29.56$ of the second sampling campaign), the model represented by Equation (1) was applied to estimate the apparent radium ages of the water samples, and the results are shown in Table 4 and Figure 10. The details are as follows.

**Table 4.** Apparent radium ages of the water samples (1 represents the first sampling, and 2 represents the second sampling). Unit: day.

| First Sampling Campaign (2 August 2017) | | |
|---|---|---|
| Sites | Equation (1) | Equation (2) |
| River1 | 0.00 | 0.00 |
| ZK1 | 25.26 | 14.91 |
| Yuanneijing1 | 20.62 | 3.74 |
| Rubiandong1 | 22.49 | 6.43 |
| Xushuichi1 | 8.12 | 1.78 |
| Kemu1 | 11.28 | 2.66 |
| Zhanggu1 | 34.94 | 15.00 |
| Mean | 17.53 | 6.36 |
| Second Sampling Campaign (16 November 2017) | | |
| Sites | Equation (1) | Equation (2) |
| River2-1 | 0.00 | 0.10 |
| River2-2 | 0.16 | 0.00 |
| ZK2 | 25.21 | 14.76 |
| Yuanneijing2 | 8.01 | 4.88 |
| Rubiandong2 | 12.09 | 4.10 |
| Xushuichi2 | 7.73 | 2.27 |
| Kemu2 | 15.48 | 7.12 |
| Zhanggu2 | 33.35 | 17.73 |
| Ganengou | 10.81 | 7.26 |
| Weishengyuan | 8.03 | 7.40 |
| Mean | 12.09 | 6.56 |

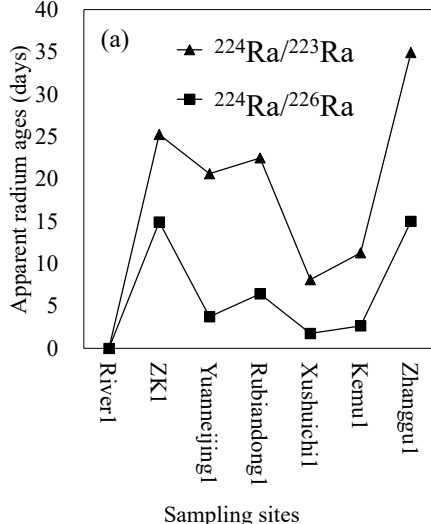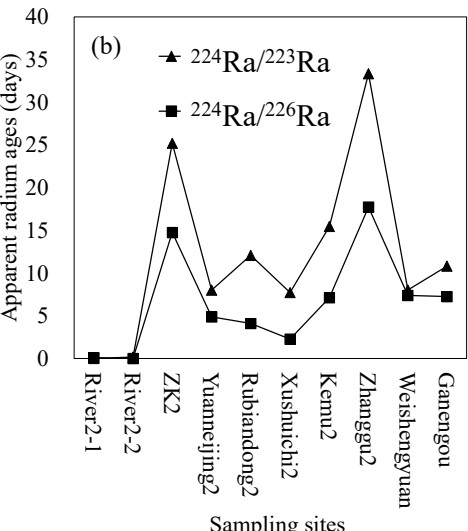

**Figure 10.** Apparent radium ages of the water samples based on the $^{224}$Ra/$^{223}$Ra radium concentration ratio (AR) and $^{224}$Ra/$^{226}$Ra AR in the (**a**) first sampling campaign and (**b**) second sampling campaign.

In the first sampling campaign (summer), the apparent radium age of Xushuichi1 (8.12 days) was the shortest time excluding River1 (0 day), followed by Yuanneijing1 (20.62 days) and Rubiandong1 (22.49 days). Xushuichi1 and Rubiandong1 were cold springs, and Yuanneijing1 was a domestic well with depth of about 10 m. The groundwater in these sites may recharged by the shallow groundwater or precipitation infiltrations through fractures and circulated at a relative shallow depth. The apparent radium age of the hot spring Zhanggu1 (34.94 days) was the longest time among the water samples, indicating that the circulation of hot springs may be at a relatively deep depth. The apparent radium age of ZK1 (25.26 days) is longer than those of the water samples circulated at a shallow depth and is shorter than that of the water samples circulated at a deep depth, suggesting that the groundwater at ZK1 may have circulated between the above two. The apparent radium age of ZK1 was long due to (1) ZK being a deep borehole with a depth of about 200 m, and (2) the velocity of groundwater at ZK being relatively slow due to the hydraulic conductivity ($3.48 \times 10^{-6}$m/s) obtained from pumping tests being similar to that of silty sand. The apparent radium age of the groundwater at Kemu1 was relatively short (11.28 days), indicating that although Kemu1 is a hot spring, it may be recharged a lot by precipitation, shallow ground or surface waters through developed fractures.

Two sampling sites of Weishengyuan and Ganengou were added to the second sampling campaign (winter). The apparent radium ages of water at Xushuichi2 (7.73 days), Yuanneijing2 (8.01 days), Weishengyuan (8.03 days), Rubiandong2 (12.09 days) and Ganengou (10.81 days) were shorter than other water samples excluding river samples (the highest AR$_{obs}$ of river samples was taken as AR$_i$, resulting in the apparent radium age of River2-1 equaling 0 days and the apparent radium age of River2-2 equaling 0.16 days), also indicating that the groundwater in these sites circulated at a relatively shallow depth. The apparent radium age of the groundwater at Zhanggu2 (33.35 days) was still the longest time because of the deep circulation. The apparent radium age of the groundwater at ZK2 (25.21 days) was still between those of the above two systems.

Similar to the results of Equation (1), the apparent radium ages estimated by Equation (2) (Table 4 and Figure 10) showed almost the same relative circulation depths among these water samples, proving the reliability of the estimated apparent radium age.

*5.4. Groundwater Flow Systems Developed in the Study Area*

The study area is located in the pull-apart basin formed by the left-lateral strike-slip movement of XF, which provides a complex fracture network system for groundwater movement. The geothermal springs develop along the regional faults. Quaternary sediment almost covers the whole study area.

The Triassic formation in the study area is thick enough (1585~4010 m) to develop different groundwater flow systems. The elevation of the study area ranges between 3000 to 4000 m. The recharge area is always located in the topographic highs, while the discharge area is located in the topographic lows. The Xianshui River is located in the regional topographic lows. Precipitation infiltrates through tectonic and/or weathering fractures, flows downward and dissolves minerals along flow paths, and the springs and mountain creeks are the forms of discharges in the study area. The groundwater flow is mainly controlled by topography, drainage patterns and fault structures. The conceptual model shown in Figure 11 describes different groundwater systems developed in the study area, showing the origin, mineralization and circulation of groundwater in this area.

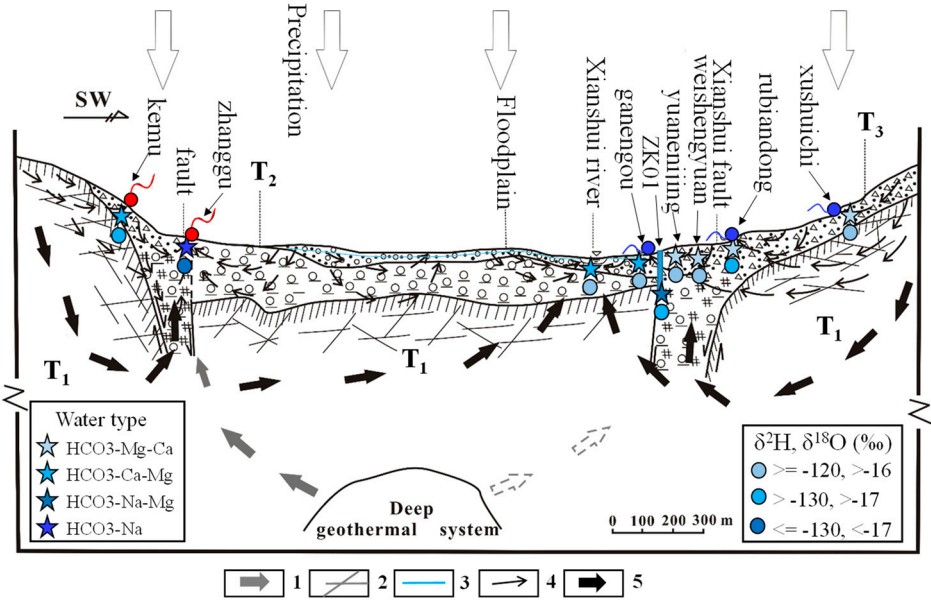

**Figure 11.** Fractured groundwater systems in the study area (note: 1. regional deep groundwater flow system (the bold grey arrows); 2. fractured and weathered zone; 3. water table; 4. local shallow groundwater system (the small thin black arrows); 5. intermediate groundwater flow system (the bold black arrows). Other symbols are the same as those in Figure 1).

Shallow groundwater (group I) is considered to belong to local groundwater systems. The groundwater in these systems is characterized by low TDS, higher $\delta^2H$ and $\delta^{18}O$ values, $HCO_3$-Ca-Mg or $HCO_3$-Mg-Ca type, and relatively short apparent radium ages. The flow system is mainly controlled by the topographic features, with recharge from the precipitation infiltrations or leakage from surface water and discharge in either topographic lows or the pumping area (e.g., cold springs and domestic wells).

The geothermal springs heated by the regional geothermal system during circulations [50] are considered to belong to regional groundwater systems (group II). The groundwater in such a system is characterized by high TDS, high temperature, lower $\delta^2H$ and $\delta^{18}O$ values, $HCO_3$-Na type, and relatively long apparent radium ages. The flow system is mainly controlled by the regional faults and other topographic features (e.g., Zhanggu and Kemu springs).

An intermediate flow system can also be delineated in the study area. The deep non-thermal groundwater (i.e., ZK, about 200 m) is characterized by the mixed hydrochemical characteristics ($HCO_3$-Na-Mg type) of other samples, increasing TDS and depleted isotopic characteristics along the intermediate flow path. The apparent radium age is between those of the above two systems. The flow system is controlled by the topographic features and the fault (e.g., ZK).

### 6. Conclusions

Groundwater circulation in the Xiashui River Fault region was investigated by collecting and analyzing the water samples in the region. Based on the chemical and isotopic compositions of the water samples, the groundwater flow systems were identified, and the following important conclusions were drawn:

(1)  Most water samples are characterized by $HCO_3$-Ca-Mg or $HCO_3$-Mg-Ca type, except for the borehole ZK which is $HCO_3$-Na-Mg type and the hot spring Zhanggu which is $HCO_3$-Na type. Chemical constituents of most water samples are mainly originated from the dissolution of carbonates, sulfates and cation exchange, while the chemical constituents of Zhanggu and ZK are mainly from the dissolution of evaporites and silicates and cation exchange.

(2)  Groundwater from the spring at Zhanggu may circulate at a deeper depth because of its high TDS and temperature as well as the high F concentration. The rest water samples excluding ZK may be circulated at a relatively shallow depth because of their similar hydrochemical compositions and lower TDS and temperature. Groundwater at ZK may be circulated at a depth between those two because of its mixed hydrochemical type, especially its concentrations of Na and Mg are between the above two groups (see Figure 2).

(3)  The three groundwater circulation depths are supported by the stable isotope data. The $^2$H and $^{18}$O values of the water samples were distributed along the local meteoric water line, inferring that groundwater in the region is recharged by precipitation. The linear regression line of $\delta^2$H and $\delta^{18}$O at hot springs Zhanggu and Kemu intercepts with the LMWL at $\delta^2$H = −145.09‰ and $\delta^{18}$O = −19.79‰. The linear regression line of $\delta^2$H and $\delta^{18}$O of the remaining water samples, excluding ZK, intercepts with the LMWL at $\delta^2$H = −93.93‰ and $\delta^{18}$O = −13.12‰. The $^2$H and $^{18}$O values of ZK are between the two intercept points.

(4)  The three groundwater circulation depths are also supported by the apparent radium age data. The hot spring at Zhanggu with high temperature has longer apparent radium age than other water samples, except for the borehole ZK, inferring that the spring water may have a longer resident time or that it is circulated at a deeper depth. The apparent radium ages of the two cold springs at Rubiandong and Ganengou, and the two domestic wells at Weishengyuan and Ganengou are similar but less than that of the hot spring at Zhanggu, indicating they may have traveled at a shorter distance or circulated at a relatively shallow depth. Groundwater at ZK has the longest apparent radium ages but may be circulated at a depth between the above two due to the relatively low hydraulic conductivity of the formation around the borehole.

(5)  Therefore, there may exist three groundwater flow systems in the study area based on the above analyses: the deep groundwater system recharged by the regional groundwater flow; the shallow groundwater system recharged by precipitations and local groundwater flow; and the intermediate groundwater system recharged by mixed local and regional groundwater flow.

Although groundwater circulation and the groundwater flow system were investigated in this study, the circulation rates of the groundwater are not known. The circulation rate of groundwater is of significant importance for the exploitation and management of water resource in this region, which requires further research from geochemical or geophysical perspectives to conduct crosschecks on the groundwater flow systems.

**Author Contributions:** Y.-K.Z., Y.Z. and other co-authors conceived the idea and designed the experiments based on ideas generated from a workshop and monthly discussions. F.L. and S.X. conducted the field experiments and situ tests in the study area. Y.Z. analyzed the test results and wrote the first draft of the paper. Y.Y. and F.L. participated in reviewing and editing the paper. All authors have read and agreed to the published version of the manuscript.

**Funding:** This research was funded by China Postdoctoral Science Foundation, grant number 2018M642897, Guangxi "Bagui Scholar" Construction Project, and Guangxi Science and Technology Planning Project, grant number GuiKE-AD18126018.

**Acknowledgments:** This study was supported by research grants from the China Postdoctoral Science Foundation (2018M642897), Guangxi "Bagui Scholar" Construction Projection, Guangxi Science and Technology Planning Project (Grant No. GuiKe-AD18126018), the Department of Science and Technology of Jiangsu Province (BE2015708), and the research fund provided by the Guangdong Provincial Key Laboratory of Soil and Groundwater Pollution Control, State Environmental Protection Key Laboratory of Integrated Surface Water-Groundwater Pollution Control, and Shenzhen Municipal Engineering Lab of Environmental IoT Technologies of the School of Environmental Sciences and Engineering, Southern University of Science and Technology, China.

**Conflicts of Interest:** The authors declare no conflict of interest.

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
