# Peer review of "Groundwater Circulation in the Xianshui River Fault Region: A Hydrogeochemical Study"

_water, doi:10.3390/w12123310_

Round 1
Reviewer 1 Report
The paper by Zhao et al. acts as a well designed hydrogeochemical study and the results obtained are of regional importance. The text is, however, quite difficult to follow because of many shortcomings and insufficient explanations for the mechanisms of groundwater cycling proposed. In my opinion, the discussion is a bit chaotic and speculative, but this is to some agree justified by the fact that there is no much data from the area of study. To come up with this problem the authors should add the outlook section to explain what is still unclear and how it is to be investigated.
Besides many specific comments to the manu evaluated (inserted to the pdf enclosed) I have picked up one important methodical aspect to clarify. Namely, the authors stabilized the water samples with HCl prior to chemical analyses. As far as I am concerned it affected HCO3- and Cl- concentrations measured. In the first case it is because pH decrease is known to influence speciation of DIC and it the latter the Cl-pool in the samples has been increased by the Cl- originating from the acid. Have the authors considered these effects?

Author Response
Comments and Suggestions for Authors
1.The paper by Zhao et al. acts as a well designed hydrogeochemical study and the results obtained are of regional importance.
Response: Thanks for your positive comments.
2.The text is, however, quite difficult to follow because of many shortcomings and insufficient explanations for the mechanisms of groundwater cycling proposed. In my opinion, the discussion is a bit chaotic and speculative, but this is to some agree justified by the fact that there is no much data from the area of study. To come up with this problem the authors should add the outlook section to explain what is still unclear and how it is to be investigated.
Response: Thanks for your suggestions. Although groundwater circulation and the groundwater flow system was investigated in this study, the circulation rates of the groundwater is not known. The circulation rate of groundwater is of significant meaning for exploitation and management of water resource in this region, which need further research from geochemical or geophysical ways to do some crosschecks on the groundwater flow systems. We added this section at the end of “Conclusion”.
3.Besides many specific comments to the manu evaluated (inserted to the pdf enclosed) I have picked up one important methodical aspect to clarify. Namely, the authors stabilized the water samples with HCl prior to chemical analyses. As far as I am concerned it affected HCO3- and Cl- concentrations measured. In the first case it is because pH decrease is known to influence speciation of DIC and it the latter the Cl-pool in the samples has been increased by the Cl- originating from the acid. Have the authors considered these effects?
Response: Sorry for the wrong description about the pre-processed method of the water samples. The fact is, water samples for chemical composition and stable isotope analyses did not do any pre-treatment, the description of “All samples were pre-processed with 0.1mol/L HCl immediately” in Line 129 is just for heavy metal ion which is beyond our study. Therefore, “All samples were pre-processed with 0.1mol/L HCl immediately, sealed, and stored in a refrigerator at 4°C until further analyzed.” in Line 129-130 was changed with “All samples were sealed with film immediately, and stored in a refrigerator at 4°C until further analyzed.”
The other specific comments to the manu has been responded and modified in the revised manu, and the details of the revisions in the manuscript and the responses to the reviewers' comments was explained point-by-point in the attachment named “Response-Comments_20201109”.

Reviewer 2 Report
The manuscript is an interesting description of the research and its results in an area whose groundwater has not been the subject of detailed research so far. Therefore, the manuscript may be of primary interest to scientists working in the region or to others as reference material.
The results are well documented by the research carried out, the discussion is interesting, and the conclusions are justified in the presented and discussed research material.
For me, the meaning of the term "apparent radium age" remains unclear. It should be explained in more detail. What do the numbers of days really mean? They are completely inconsistent with the age of the water - the time of its underground flow. So what do they mean? On the basis of the value of hydraulic conductivity for the ZK well, it can be quickly calculated that the time of water inflow to the bottom of the well from the river will be approximately 2-3 years. On the other hand, the values of "apparent radium age" are expressed in days and have nothing to do with this value. This requires a more extensive explanation.
Besides, I have a few minor comments:
- The term "radium quartet" is not a scientific concept, so I propose either to explain it or simply use the term "four natural radium isotopes" instead.
- Why do authors use the old dpm (decay per minute) unit instead of the current unit - becquerel. In my opinion, concentrations of Ra isotopes activity should be expressed in Bq / L.
- In Figure 1, not for all readers the marks T3 and T2 (numbers 18 and 19) may be clear. They need maybe additional explanation. Insted of Zhang, 1988 here should be placed the right number of reference. This publication should be inserted in references if it is not done yet.
- Line 126: Here the new sentence should starts from T (instead of t).
- In Figure 4, units are missing on the axes of the charts. In Figure 4a, not only the 223Ra isotope should be described on the y axis, but all Ra isotopes.
In my opinion, it is also worth re-checking the grammatical correctness of a few sentences in the first three chapters of the manuscript.
Reviewer 3 Report
Dear Autors
The article is very interesting. I don't have the significant comments to the article. In my opinion, it should be published in the presented version.
Kind regards,
Round 2
Reviewer 1 Report
Actually, I do not havr any further comments. The changes suggested have been implemented. As far as I am concerned the study is definitely publishable now and it is of potential interest for a broad community of hyrogeologists and hydrogeochemists.